# Node-wise Calibration of Graph Neural Networks under Out-of-Distribution Nodes via Reinforcement Learning

## Abstract

Graph neural networks (GNNs) achieve great success in tasks like node classification, link prediction, and graph classification. The core of GNNs aims to obtain representative features by aggregating neighborhood node information through the message-passing mechanism. However, when the graph is mixed with out-of-distribution (OOD) nodes, existing methods generally fail to provide reliable confidence for in-distribution (ID) classification, due to the under-explored negative impact from the OOD nodes. Our studies suggest that the calibration issue of GNN with OOD nodes is more complicated than that without OOD nodes. In some datasets the predictions of GNN are under-confident issue while others may be over-confident. This irregularity makes the current calibration methods less effective since none of them considers the negative impact from OOD nodes. Inspired by the existing work that calibrates the neural network with new loss functions that aim to adjust the entropy of the output implicitly, we aim to achieve the same goal by adjusting the weight of the edges. Our empirical studies suggest that manually lowering the weight of edges connecting ID nodes and OOD nodes could effectively mitigate the calibration issue. However, identification of these edges and determination of their weights remains challenging since the OOD nodes are unknown to the training process. To tackle the above challenge, we propose a novel framework called RL-enhanced Node-wise Graph Edge Re-weighting (RNGER) to calibrate GNNs against OOD nodes. The proposed RNGER framework explores how the entropy of the target nodes is affected by the adjustment of the edge weights without the need for identifying OOD nodes. We develop the iterative edge sampling and re-weighting method accordingly and formulate it as the Markov Decision Process. With the reinforcement learning method, we could achieve the optimal graph structure to alleviate the calibration issue of GNNs. Experimental results on benchmark datasets demonstrate that our method can significantly reduce the expected calibration error (ECE) and also show comparable accuracy, compared with strong baselines and other state-of-the-art methods.

## 1 Introduction

Graph-structured data are prevalent in the real world, such as social networks, traffic networks, and biological molecules. To deal with graph-structured data, graph neural networks (GNNs) have recently been the mainstream backbones, which model the representative features of nodes by aggregating the information from neighbors. However, the reliability of GNN predictions is an issue worthy of discussion that is under-explored, especially for safety-critical applications. A previous study (Guo et al., 2017) proposes the expected calibration error (ECE) to measure the difference between the confidence of the prediction and the accuracy yielded by the neural network. The latest work (Wang et al., 2021b; Hsu et al., 2022a; Teixeira et al., 2019) also points out that the GNNs could yield prediction results with large calibration errors.

Up to now, several work has been done to tackle the calibration issue of (graph) neural networks. One line of the work (Guo et al., 2017; Zadrozny & Elkan, 2001; Gupta et al., 2020; Wang et al., 2021b; Zhang et al., 2020; Hsu et al., 2022a) aims to calibrate the neural network with post-hoc method. Another branch of work (Mukhoti et al., 2020; Ghosh et al., 2022; Tao et al., 2023; Wang et al., 2022)

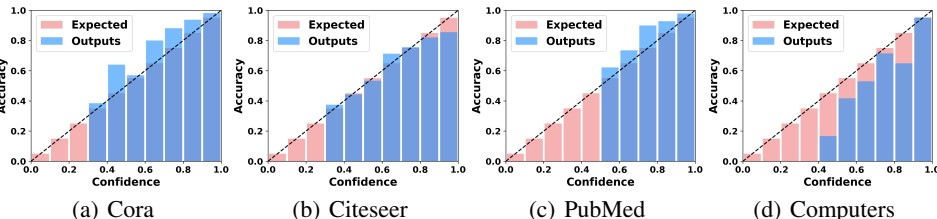

Figure 1: Reliability diagrams of GCN on (a) Cora, (b) Citeseer, (c) PubMed and (d) Amazon_Computers with OOD nodes. Well-calibrated results would have closer alignment with the expected results along the diagonal line. The results suggest that the calibration issue is different and complicated on different datasets.

addresses the calibration issue by adopting new functions in the training such as focal loss (Lin et al., 2017). This line of work implies that the new loss function can implicitly adjust the entropy of the output from neural networks and therefore can calibrate the logits of neural networks.

Existing GNNs deal with graph-structured data in which all nodes are in-distribution (ID) nodes. However, in the real world, graphs are often comprised of a large number of out-of-distribution (OOD) nodes (Stadler et al., 2021; Zhao et al., 2020; Yang et al., 2022; Song & Wang, 2022). For instance, users on social networks are usually linked with strangers and online scammers apart from their family members and friends. In financial transaction networks, there are plenty of financial fraudsters connected to normal users. When a graph is mixed with OOD nodes, the calibration issue is more complicated. As shown in Fig 1, unlike the general under-confidence problem of GNNs (Wang et al., 2021b; Hsu et al., 2022a), on some datasets, the results of GNNs are over-confident and others may experience the under-confidence problem. Our experiments also demonstrate that the existing calibration method would be less effective on the graph with OOD nodes since these methods don't consider the negative impact of OOD nodes.

Our empirical studies suggest that by manually lowering the weight of edges that are connecting to OOD nodes, the calibration issue can be mitigated to some extent. However, identifying the OOD nodes in the graph is not a trivial problem. To this end, we propose an RL-enhanced Node-wise Graph Edge Re-weighting framework called RL-enhanced Node-wise Graph Edge Re-weighting (RNGER) method to calibrate GNNs without explicitly the need to identify the OOD nodes. Our method conforms to the Actor-Critic paradigm. Inspired by the previous work (Mukhoti et al., 2020; Ghosh et al., 2022; Tao et al., 2023; Wang et al., 2022) that calibrates the output logits of neural works by implicitly adjustment of the entropy, we intend to perform the same task through new weight of edges learned from deep deterministic policy gradient (DDPG) (Lillicrap et al., 2016) method. In our method, we sample the labeled nodes as well as the neighborhood edges. Then the iteration of the neighborhood edges would be formulated as a Markov Decision Process (MDP) in our method and the weight of edges would be adjusted dynamically to evaluate the change of the entropy for each of the sampled nodes. The new reward signal is designed to direct the Actor network to yield a new weight of edges that can enlarge (lower) the entropy for the target sampled node if it is over-confident (under-confident). Through the reinforcement learning, the optimal graph structure could be obtained and the calibration issue on the noisy graph can be mitigated. The contribution of this paper is summarized as follows:

- We propose RL-enhanced Node-wise Graph Edge Re-weighting (RNGER) framework to calibrate graph neural networks when the graph is mixed with OOD nodes. We develop an iterative edge sampling and re-weighting scheme and formulate it as the Markov Decision Process (MDP). A new reward is designed to guide the training of our framework.

- Existing GNN can be incorporated into our framework. With the modified edge weights, our method can yield lower calibrate error as well as comparable accuracy compared to the state-of-the-art methods.

- Experimental results further show that the learned edge weights are transferable and can be beneficial in graph learning with other GNN methods. The performance would be improved in some tasks, such as node classification and OOD detection.

## 2 RELATED WORK

**Neural Network Calibration.** The pursuit of developing a reliable and trustworthy model has captured the attention of researchers, leading to its extension into the realm of graph neural networks. Guo et al. (Guo et al., 2017) first proposed the calibration error to measure the confidence of the results from deep neural networks. Extensive work (Mukhoti et al., 2020; Ghosh et al., 2022; Tao et al., 2023; Wang et al., 2022) has been done on the calibration of neural networks. Recent work (Wang et al., 2021b) post-processes the logits of the GCN (Kipf & Welling, 2016) model to obtain the calibrated results. Uncertainty estimation (Lakshminarayanan et al., 2017; Malinin & Gales, 2018) also benefits the network calibration by modeling the probability distribution of the predicted labels. Wang et al. (Wang et al., 2022) proposed GCL loss to mitigate the under-confidence issue of GNNs in an end-to-end manner. Recently, GATS (Hsu et al., 2022a) is designed to account for the influential factors that affect the calibration of GNN.

**Reinforcement Learning on Graph.** The rapid development of Reinforcement Learning (RL) in cross-disciplinary domains has motivated scholars to explore novel RL models to address graph-related problems, such as neighborhood detection, information aggregation, and adversarial attacks. GraphNAS (Gao et al., 2019) designs a search space covering sampling functions, aggregation functions, gated functions and searches the graph neural architectures with RL. Policy-GNN (Lai et al., 2020) adaptively determines the number of aggregations for each node via deep Q-learning (Mnih et al., 2013). RL-Explainer (Shan et al., 2021) and GFlowExplainer (Li et al., 2023) adopt off-policy RL methods for graph explanation.

**Graph Learning with OOD.** Most graph learning is built on the hypothesis that training and testing data are independent and identically distributed (I.I.D.). Song et al. (Song & Wang, 2022) first proposed graph learning with OOD nodes and develop OODGAT (Song & Wang, 2022) framework to perform both the node classification and OOD nodes detection. The core of the OODGAT (Song & Wang, 2022) is to identify the OOD nodes and reduce the connection between ID nodes and OOD nodes. Another line of work focus on the graph OOD detection. GNNSAGE (Wu et al., 2023) performs OOD node detection by a learning-free energy belief propagation scheme. In GPN (Stadler et al., 2021) OOD nodes detection is completed by the uncertainty estimation. GraphDE (Li et al., 2022), a probabilistic generative framework, can jointly perform graph debiased learning and out-of-distribution nodes detection.

## 3 BACKGROUND

### 3.1 PROBLEM FORMULATION

We first present the problem formulation of our study. Consider an attributed graph $\mathcal{G} = \{\mathcal{V}, \mathcal{E}, X\}$ where the finite node set is denoted by $V = \{i|1 \leq i \leq N\}$, and the edge set is denoted by $\mathcal{E} \subseteq \mathcal{V} \times \mathcal{V}$. $N$ is the total number of the nodes in the graph, and the feature matrix is denoted by $\mathbf{X} \in \mathbb{R}^{N \times d}$ in which $d$ is the length of the feature vector. The structure of the graph $\mathcal{G}$ can be represented by the binary adjacency matrix $A = \{0, 1\}^{N \times N}$. In graph learning with out-of-distribution (OOD) nodes, The nodes set can be split into an ID node set and an OOD node set $\mathcal{V} = \mathcal{V}_{ID} \cap \mathcal{V}_{OOD}$. The feature of OOD nodes is sampled from a different distribution than that of ID nodes, i.e., $P(X_{OOD}) \neq P(X_{ID})$. The label space for the ID node set is $Y = \{1, 2, \cdots, K\}$, while we assume that the OOD nodes do not fall into any existing category of the ID nodes, and their labels are unknown to us. In semi-supervised graph learning, the ID nodes can be further divided into labeled ID nodes and unlabelled ID nodes, i.e., $\mathcal{V}_{ID} = \mathcal{V}_{ID}^l \cap \mathcal{V}_{ID}^{ul}$. The goal of standard semi-supervised graph learning is to learn a classifier $f : X, A \to \tilde{Y}$ that maps the feature of the nodes and graph structural information to the predicted labels $\tilde{Y}$ of the nodes. As aforementioned, the task becomes more challenging with the presence of unknown OOD nodes. How to rule out the negative impact from the OOD nodes is the key for the semi-supervised graph learning with OOD nodes.

In our study, the expected calibration error (ECE) is considered as a major metric. According to the practice in related work (Guo et al., 2017), the predictions are regrouped into $M$ equally spaced confidence intervals $(B_1, B_2, \cdots, B_M)$ with $B_m = \{i \in \mathcal{V}|\frac{m-1}{M} < \tilde{p}_i \leq \frac{m}{M}\}$ where $\tilde{p}_i$ is the

Table 1: Comparison between GCN with original and modified edge weights in terms of node classification accuracy (Acc%) and expected calibration error (ECE%). The experiments are repeated 10 times and the average results are reported. The bold represents the best results.

| Edge weight | Cora | | Citeseer | | PubMed | | Photo | | Computers | | Arxiv | |
|---|---|---|---|---|---|---|---|---|---|---|---|---|
| | Acc | ECE | Acc | ECE | Acc | ECE | Acc | ECE | Acc | ECE | Acc | ECE |
| Original | **86.26** | 6.64 | 70.41 | 4.81 | **92.09** | 1.22 | **93.30** | 3.14 | **88.23** | 6.45 | 80.38 | 5.02 |
| Modified | 85.94 | **6.01** | **70.75** | **4.41** | **92.09** | **1.12** | 93.27 | **2.63** | 88.07 | **5.85** | **80.60** | **4.24** |

confidence for node $i$. And the expected calibrated error (ECE) can be defined as:

$$\text{ECE} = \sum_{m=1}^{M} \frac{|B_m|}{|\mathcal{V}|} |\text{acc}(B_m) - \text{conf}(B_m)|, \tag{1}$$

where

$$\text{acc}(B_m) = \frac{1}{|B_m|} \sum_{i \in B_m} \mathbb{1}(\tilde{y}_i = y_i) \quad \text{and} \quad \text{conf}(B_m) = \frac{1}{|B_m|} \sum_{i \in B_m} \tilde{p}_i. \tag{2}$$

## 3.2 DEEP REINFORCEMENT LEARNING

Reinforcement learning plays an important role in the decision making process, and one representative method is the Markov Decision Process (MDP). A typical MDP can be formulated as $\mathcal{M} = \{\mathcal{S}, \mathcal{A}, P_\pi, r, \gamma\}, \rho_0$, where $\mathcal{S}$ is the state space, $\mathcal{A}$ is the action space, $P_\pi(s'|s, a)$ is the state-action transition probability, $r$ is the reward function, $r$ is the reward function, $\gamma \in (0, 1)$ is the discount factor ,and $\rho_0$ is the initial state distribution over state space $\mathcal{S}$. The goal of off-policy reinforcement learning is to learn the policy $\pi(a|s)$ that can maximize the discounted cumulative reward $J_\pi = \sum_{t=0}^{\infty} \gamma^t r(s_t, a_t)$ by training on the outcomes produced by a different behavior policy rather than that produced by the target policy. One of the most well-known off-policy method in deep learning is deep Q-learning (Mnih et al., 2013; Van Hasselt et al., 2016). The basic idea of deep Q-learning is to approximate the Q function by deep neural networks, and the policy is obtained from the estimated value of $a = \text{argmax}_a Q(s, a) = \text{argmax}_a \mathbb{E}_{s' \sim \mathcal{S}}(r + \gamma \max_{a'} Q(s', a'))$.

Apart from Q-value based methods that obtain the action implicitly from the Q function, policy gradient methods (Haarnoja et al., 2018; Wang et al., 2017; Cobbe et al., 2021; Barth-Maron et al., 2018; Tkachenko, 2015; Silver et al., 2014b; Mnih et al., 2016) instead aim to learn the policy directly by parameterized function $\pi_\theta(a)$. Similar to deep Q-learning (Mnih et al., 2013; Van Hasselt et al., 2016), we update the parameter $\theta$ in the policy function to achieve the maximum discounted cumulative reward. Besides, modern off-policy gradient methods (Haarnoja et al., 2018; Wang et al., 2017; Cobbe et al., 2021; Barth-Maron et al., 2018; Tkachenko, 2015) adopt the actor-critic algorithm that models the policy and Q function to achieve better learning efficiency and convergence. The parameter $\theta$ of policy function can be updated according to the Policy Gradient Theorem (Sutton et al., 1999):

$$\nabla_\theta J(\theta) = \mathbb{E}_\pi[\nabla \ln \pi(a|s, \theta) Q_\pi(s, a)]. \tag{3}$$

## 4 EMPIRICAL STUDY

In this section we aim to investigate if the calibration error of GNNs can be reduced with adjusted edge weights when a graph is mixed with OOD nodes, We follow the previous work (Zhao et al., 2020; Stadler et al., 2021) to divide the the existing nodes into ID nodes and OOD nodes and choose GCN (Kipf & Welling, 2016) as the target model. Suppose the labels and distribution of all nodes is known and we manually modify the weight of edges (e.g., 1 to 0.5) that are connecting to OOD nodes. The experiments are evaluated on the six benchmark datasets. The details of the benchmark can be found in Table 2. The results in Table 1 suggest that lowering the weight of corresponding edges can definitely reduce the calibration error while maintaining comparable accuracy on node classification compared to that with original edge weights. Based on these findings, we are motivated to develop new methods to obtain new edge weights to calibrate graph neural networks.

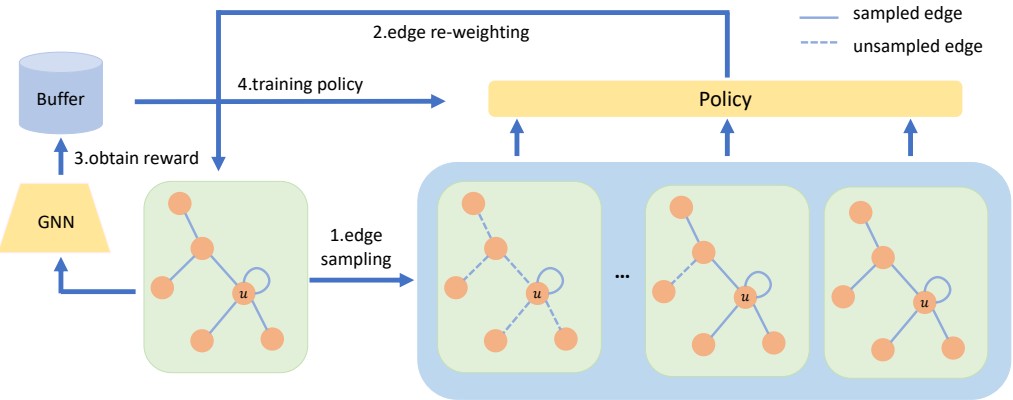

Figure 2: The illustration of our RL-enhanced Node-wise Graph Edge Re-weighting (RNGER) framework. The method consists of four steps. In the first step, we iteratively traverse the adjacent edges. In the beginning, only self-loop edge is taken into consideration. Each time we sample a new edge within the subgraph without replacement and form the state. In the second step, the adjusted weight would be obtained from the state and assigned to the new sampled edge. Next, reward $r$ is obtained from the GNN backbone with adjusted edge weight, and the transition tuple is stored in the replay buffer. Finally, we adopt the DDPG method to train our policy function.

## 5 METHODOLOGY

In this section, we give an overview of our framework. As aforementioned, our method is motivated by previous work (Mukhoti et al., 2020; Ghosh et al., 2022; Tao et al., 2023; Wang et al., 2022) that calibrates the (graph) neural network by implicitly regulation of the entropy and our empirical studies. In this section, we first introduce the formulation of our edge iteration process and how we incorporate DDPG (Lillicrap et al., 2016) the generate new edge weights to regularize the entropy of the nodes under the guidance of the reward signal. Then we provide details of our whole method. Besides, we also provide some discussions on the justification and time complexity analysis.

### 5.1 ITERATIVE EDGE SAMPLING AND RE-WEIGHTING

For a target node $u$, the edge set within the 2-hop subgraph is denoted as $\mathcal{E}^u = \{e_0^u, e_1^u, \cdots, e_m^u\}$. Specifically, $e_0^u$ is the self-loop edge for node $u$. To quantitatively evaluate the impact of the edge re-weighting for the target node, we sequentially sample the edges without replacement from $\mathcal{E}^u$ and modify their edge weights. Specifically, At time $t = 0$, we only consider the re-weighting of the self-loop edge $e_0^u$. From time $t = m - 1$ to $t = m$, a new edge (i.e., $e_m^u$) is sampled from $\mathcal{E}^u$, and the edge weight is adjusted accordingly. In the whole time, the weights of unsampled edges remain the same. Since the iterative edge sampling and the re-weighting process are formulated as a Markov Decision Process in our framework, we provide the definitions of state, action, and reward as follows.

**State**. The state $s_t \in \mathcal{S}$ at timestamp $t$ in our framework is defined as:

$$s_t = h(s_{t-1}, f_{e_t}), \tag{4}$$

where $f_e$ is the feature of the edge $e$, and $h$ is the function that maps the old state and new edge feature into a new state. We adopt the average of the features of the connecting nodes as the edge feature. At time $t = 0$, $s_0 = X_u$. In our study, we adopt the moving average method to generate the state:

$$s_t = \alpha f_{e_t} + (1 - \alpha)s_{t-1}, \tag{5}$$

where $\alpha$ is the hyper-parameter that balances the contribution of new edge features in the state.

**Action**. In our method, the action $a \in \mathcal{A}$ we take for each new sampled edge is to adjust its weight. Since in our case, the action space is continuous $\mathcal{A} \subseteq (0, 1]$, we adopt the policy function to generate the adjusted edge weight from the state $s$. At time $t$, the edge weight $w_{e_t}$ for $e_t$ is generated by:

$$w_{e_t} = \pi(s_t|\theta^\pi), \tag{6}$$

where $\pi_\theta$ is the policy function which can be implemented as a neural network with the Sigmoid activation function in the last layer to ensure the output is between $0$ and $1$.

**Reward**. The reward signal $r$ is designed to encourage the policy function to produce new edge weights, in order to enlarge (lower) the entropy of the target nodes. To determine if the node is over-confident or under-confident, we evaluate the calibration error on the validation nodes and obtain the $\mathrm{acc}(B_m)$ and $\mathrm{conf}(B_m)$ illustrated in Eq. equation 2 for each bin during training. If the predictive probability of the target node falls into certain bin $m$, then the reward would be defined as:

$$r(s, a) = \mathbb{1}(\tilde{y}_i = y_i) + \beta H_i \quad \text{where} \quad \beta = \begin{cases} +1 & \text{if} \quad \mathrm{acc}(B_m) - \mathrm{conf}(B_m) < 0 \\ -1 & \text{if} \quad \mathrm{acc}(B_m) - \mathrm{conf}(B_m) > 0 \end{cases}, \quad (7)$$

where $\tilde{y}_i$ is the predicted label for node $i$ generated by the GNN backbone, and $y_i$ is the ground truth label. $H$ is the entropy, and $\beta$ is the coefficient that determines the sign of entropy in the reward according to whether the validation nodes in bin $m$ are over-confident or under-confident. In Eq. equation 7, the first term can be regarded as the accuracy on the ID nodes. The second term aims to regularize the entropy of the target node based on its own situation.

## 5.2 Details of Algorithm

The framework of our proposed method is illustrated in Fig. 2. The framework basically consists of four steps. In this first step, we form the candidate node set $\mathcal{I}$ from the training and validation nodes. For each candidate node, we iteratively sample the adjacent edges and form the state, as discussed in Sec. 5.1. In step two, the adjusted edge weight is obtained from the policy function $\pi_\theta(s)$. In order to enhance the exploration ability of the policy function in the continuous action space, we reformulate our adjusted edge weight as:

$$w_{e_t}^* = \pi(s_t|\theta^\pi) + \epsilon, \quad (8)$$

where $\epsilon$ is the noise following a Gaussian distribution $\epsilon \sim \mathcal{N}(0, \sigma)$. The $\sigma$ changes with iteration:

$$\sigma = \sigma_0 (1 + \frac{t}{T})^{-d}, \quad (9)$$

where $\sigma_0$ is the initial value of noise, $T$ is the total number of iterations. $d > 0$ is the decay rate. In the next step we obtain the reward $r$ from the GNN backbone according to Eq. equation 7 and the tuple of transition $(s_t, a_t, r_t, s_{t+1})$ is stored in the replay buffer $\mathcal{B}$. In the final step, we adopt the deep deterministic policy gradient (DDPG) (Lillicrap et al., 2016) method to train our policy function, because the state and action spaces are all continuous in our problem. DDPG (Lillicrap et al., 2016) adopts the actor-critic framework for better stability and convergence of the training. Similar to deep Q-learning (Mnih et al., 2013), the objective of critic network $Q(s_t, a_t|\theta^Q)$ is to approximate the discounted cumulative reward from the state-action pair by minimizing the loss:

$$L(\theta^Q) = \mathbb{E}_{s_t \sim \mathcal{S}, a_t \sim \mathcal{A}}[(Q(s_t, a_t)|\theta^Q) - y_t)^2], \quad (10)$$

where $y_t$ can be derived from the Bellman equation (Sutton & Barto, 2018):

$$y_t = r(s_t, a_t) + \gamma Q(s_{t+1}, \pi(s_{t+1}|\theta^\pi)|\theta^Q). \quad (11)$$

Since our policy function yields the continuous edge weight deterministically from the state, the parameter of policy can be updated according to the Deterministic Policy Gradient Theorem (Silver et al., 2014a; Lillicrap et al., 2016):

$$\nabla_{\theta^\pi} J = \mathbb{E}_{s_t}[\nabla_a Q(s, a|\theta^Q)_{s=s_t, a=\pi(s_t|\theta^\pi)} \nabla_{\theta^\pi} \pi(s|\theta^\pi)_{s=s_t}]$$
$$\approx \frac{1}{N} \sum_i (\nabla_a Q(s, a|\theta^Q)_{s=s_i, a=\pi(s_i|\theta^\pi)} \nabla_{\theta^\pi} \pi(s|\theta^\pi)_{s=s_i}). \quad (12)$$

The detailed procedures of our proposed method are summarized in Algorithm 1.

## 5.3 Analysis

As aforementioned, the current calibration methods (Mukhoti et al., 2020; Ghosh et al., 2022; Tao et al., 2023; Wang et al., 2022) adopt new loss function for training neural network. For instance, focal loss $\mathcal{L}_{FL} = -(1-\hat{p})^\gamma \log \hat{p}$ (Lin et al., 2017) and inverse focal loss $\mathcal{L}_{inv\_FL} = -(1+\hat{p})^\gamma \log \hat{p}$ (Wang et al., 2021a) have been adopted to calibrate the over-confident and under-confident output of neural networks, respectively. Both functions can achieve better calibrated result by regularizing the entropy implicitly.

---

**Algorithm 1** Algorithm of our RNGER framework

---

**Input:** input graph $\mathcal{G} = (\mathcal{V}, \mathcal{E}, X)$, GNN backbone $f$, labels of the nodes $\mathbf{Y}$, candidate nodes set $\mathcal{I}$, critic network $Q(s, a|\theta^Q)$, actor network $\pi(s|\theta^\pi)$, replay buffer $\mathcal{B}$, discount coefficient $\gamma$, hyperparameter $\alpha$, initial noise $\sigma_0$, the total episode $P$., adjacent matrix $A$.

Initialize the actor network $\pi$, critic network $Q$ and replay buffer $\mathcal{B}$.

**for** 1,2,3..., P **do**

    train the GNN backbone $f$ with adjacent matrix $A$ and obtain the $\text{acc}(B_m)$ and $\text{conf}(B_m)$ on validation nodes. Sample one target node $u$ from the candidate nodes set $\mathcal{I}$.

    obtain the edge set $\mathcal{E}^u = \{e_0^u, e_1^u, \cdots, e_m^u\}$ for target node $u$.

    **for** $e_t$ in $\mathcal{E}^u$ **do**

        obtain the state $s_t$ by Eq. equation 5.

        calculate the adjusted edge weight from state $s_t$ by Eq. equation 6.

        add the noise to the adjusted edge weight for exploration via Eq. equation 8 and Eq.equation 9.

        assign the adjusted edge weight to the original graph $\mathcal{G}$.

        obtain the reward $r$ from the GNN backbone $f$ via Eq. equation 7.

        form the transition tuple $(s_t, a_t, s_{t+1}, r_t)$ into replay buffer $\mathcal{B}$.

        randomly sample the data from replay buffer $\mathcal{B}$ and train the actor network $\pi$ and critic network $Q$ via Eq. equation 10 and Eq equation 12.

    **end for**

    generate the new edge weights and obtain new adjacent matrix $A'$ using Eq. equation 6. Train the GNN backbone $f$ and save the actor and critic networks based on the evaluation of model $f$.

    update the adjacent matrix $A = A'$

**end for**

---

**Proposition 1** *The focal loss $\mathcal{L}_{FL}$ is the upper bound on the regularised KL-divergence of target distribution $q$ and the predicted distribution $\hat{p}$, where the regulariser is the negative entropy of the predicted distribution $\hat{p}$. $\mathcal{L}_{FL} \geq KL(q||\hat{p}) - \gamma H(\hat{p})$.*

**Proposition 2** *The inverse focal loss $\mathcal{L}_{inv_F L}$ is the lower bound on the regularised KL-divergence of target distribution $q$ and the predicted distribution $\hat{p}$, where the regulariser is the negative entropy of the predicted distribution $\hat{p}$. $\mathcal{L}_{inv\_FL} \leq KL(q||\hat{p}) + \gamma H(\hat{p})$.*

As suggested by Proposition 1 and Proposition 2, the over-confidence (under-confidence ) issue can be alleviated by enlarging (lowering) the output entropy. As for graph with OOD nodes, the calibration problem of GNNs varies on different datasets and no single loss function can be applied to all datasets. Thus we consider regularizing the entropy through our modified edge weights obtained by reinforcement learning.

As for time complexity, suppose the $L$ is the number of layers in GCN, $|E|$ is the number of edges, $|N|$ is the number of nodes, $|F|$ is the dimension of the features, $N_d$ is the number of target nodes, $d$ is the average number of edges within 2-hop. $h$ is the hidden dimension of the three-layered actor neural network. The time complexity is $O(N_d * (L|E|F+)d + LNF^2 + Fh)$. The time complexity depends on the number of target nodes and the average number of adjacent edges. For large graphs, we can choose a proper value to maintain the reasonable time cost.

## 6 EXPERIMENTS

In this section, we first introduce the experimental settings in our study. Then we show the main results of the experiment as well as the visualization of the reliability diagram and distribution of the edge weights. The ablation study is focused on the investigation of the components of the reward on the performance of our framework. Finally, we show a case study on the benefit of our learned edge weights on graph learning with other methods.

Table 2: The statistics of datasets

| Dataset | ID classes | OOD classes | #Nodes | #Edges | #Features | #Classes |
|---|---|---|---|---|---|---|
| Cora | [0 - 3] | [4 - 6] | 2,708 | 10,556 | 1,433 | 7 |
| Citeseer | [0 - 2] | [3 - 5] | 3,327 | 9,104 | 3,703 | 6 |
| PubMed | [0 - 1] | [2] | 19,717 | 88,648 | 500 | 3 |
| Amazon-Photo | [0 - 3] | [4 - 7] | 7,650 | 238,162 | 745 | 8 |
| Amazon-Computers | [0 - 4] | [5 - 9] | 13,752 | 491,722 | 767 | 10 |
| OGB-Arxiv | [0 - 1] | [19 - 39] | 169,343 | 1,166,243 | 128 | 40 |

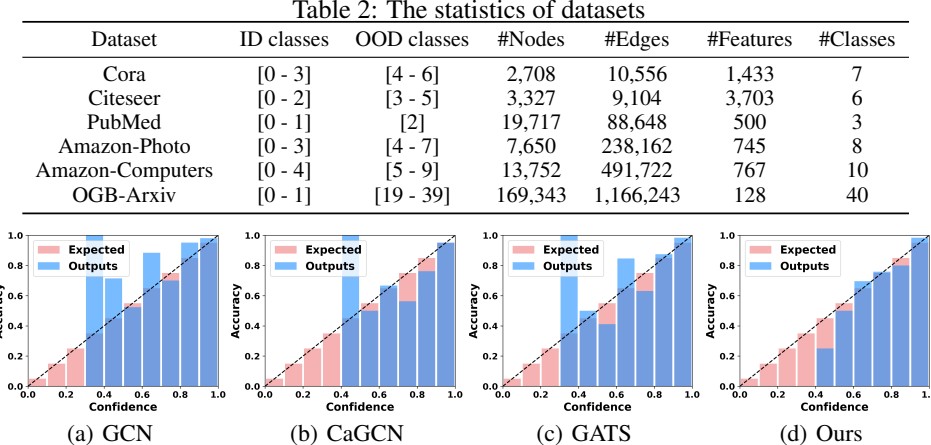

(a) GCN    (b) CaGCN    (c) GATS    (d) Ours

Figure 3: Reliability diagrams of (a) GCN, (b) CaGCN, (c) GATS and (d) our proposed method on Amazon-Photo. Well-calibrated results would have closer alignment with the expected results along the diagonal line.

## 6.1 EXPERIMENTAL SETTINGS

In the experiments, we perform the semi-supervised node classification task and compare the performance of our framework with the baseline methods on six benchmark datasets. The ablation study and case study can be found in appendix.

**Datasets**. We adopt six public benchmark datasets, including Cora, Citeseer, PubMed (Yang et al., 2016), Amazon-Photo, Amazon-Computers (Shchur et al., 2018) and OGB-Arxiv (Hu et al., 2020), for evaluating our method and baselines. We basically adhere to the train/validation/test splits provided by previous work (Yang et al., 2016; Shchur et al., 2018). To formulate the graph learning with the OOD nodes setting, we manually split the nodes into ID nodes and OOD nodes. For instance, Cora (Yang et al., 2016) has 7 classes and the nodes from the first 4 classes would be regarded as ID nodes. The rest are OOD nodes and would be marked out in the training and validation. More details of the datasets are illustrated in Table 2.

**Baselines**. The baselines include GCN (Kipf & Welling, 2016), HyperU-GCN (Yang et al., 2022), CaGCN (Wang et al., 2021b), GATS (Hsu et al., 2022b), GCL (Wang et al., 2022) and OODGAT (Song & Wang, 2022). More details can be found in appendix.

**Metrics**. In our experiments, we adopt the expected calibration error (ECE) (Guo et al., 2017) as our major metric. The lower value of ECE means the better reliability of the prediction results from GNN models. Besides, we also report the node classification accuracy.

**Implementation Details**. In our method, we adopt GCN (Kipf & Welling, 2016) and HyperU-GCN (Yang et al., 2022) as our GNN backbone. The hyper-parameters of GCN are the same as the corresponding baselines. The learning rate is 1e-2 and weight decay is 5e-4. The hidden dimension is 128. The Actor and Critic in our framework are implemented as a three-layered MLP with the dimension of hidden layers 256 and 16, respectively. More details can be found in appendix.

## 6.2 EXPERIMENTAL RESULTS AND VISUALIZATION

Table 3 and Table 4 show the performance of our proposed method and the baselines on the benchmarks. The results show that the ordinary GNN models such as GCN (Kipf & Welling, 2016) would yield large calibration errors. For instance, the ECE can reach $6.64\%$ on Cora. Besides, the results also suggest that the methods aimed at the calibration of GNNs also experience large calibration errors on some datasets. For instance, though CaGCN (Wang et al., 2021b) can achieve lowest calibration error on Cora (Yang et al., 2016) and Amazon-Computers (Shchur et al., 2018), the calibration error would reach 15%. The cause of the phenomenon can be attributed to the adverse impact of the OOD nodes on the ID nodes and make the regularization term in CaGCN (Shchur et al., 2018) less effective on large dataset. GCL (Wang et al., 2022) can achieve better calibration error than that of GCN (Kipf

Table 3: Comparison between our proposed method and other baselines in terms of node classification accuracy (Acc%) and expected calibration error (ECE%) on Cora, Citeseer and PubMed. The experiments are repeated 10 times and the average results and standard deviation are reported. The bold represents the best results.

| Methods | Cora | | Citeseer | | PubMed | |
|---|---|---|---|---|---|---|
| | Acc | ECE | Acc | ECE | Acc | ECE |
| GCN (Kipf & Welling, 2016) | $86.26 \pm 0.45$ | $6.64 \pm 0.19$ | $70.41 \pm 0.67$ | $4.81 \pm 0.26$ | $92.09 \pm 0.22$ | $1.22 \pm 0.19$ |
| CaGCN (Wang et al., 2021b) | $86.99 \pm 0.26$ | $\mathbf{2.50} \pm 0.20$ | $71.30 \pm 0.57$ | $4.09 \pm 1.17$ | $92.34 \pm 0.20$ | $2.68 \pm 0.19$ |
| GATS (Hsu et al., 2022b) | $\mathbf{87.06} \pm 0.18$ | $2.63 \pm 0.66$ | $\mathbf{71.95} \pm 0.43$ | $\mathbf{4.03} \pm 1.46$ | $\mathbf{92.69} \pm 0.26$ | $1.96 \pm 0.27$ |
| GCL (Wang et al., 2022) | $86.32 \pm 0.38$ | $6.42 \pm 0.19$ | $70.36 \pm 0.67$ | $4.56 \pm 0.78$ | $92.01 \pm 0.21$ | $1.16 \pm 0.35$ |
| OODGAT (Song & Wang, 2022) | $83.65 \pm 1.66$ | $6.60 \pm 3.81$ | $61.82 \pm 0.92$ | $8.86 \pm 1.40$ | $87.44 \pm 0.91$ | $4.64 \pm 1.28$ |
| HyperU-GCN (Yang et al., 2022) | $85.47 \pm 0.98$ | $7.96 \pm 9.94$ | $70.86 \pm 2.34$ | $22.37 \pm 11.11$ | $91.20 \pm 0.63$ | $2.32 \pm 0.41$ |
| RNGER+GCN (Ours) | $85.54 \pm 0.53$ | $6.43 \pm 0.26$ | $70.61 \pm 0.83$ | $4.63 \pm 0.99$ | $91.79 \pm 0.21$ | $\mathbf{1.07} \pm 0.33$ |
| RNGER+GATS (Ours) | $86.81 \pm 0.29$ | $3.02 \pm 0.67$ | $71.72 \pm 0.72$ | $4.23 \pm 1.10$ | $92.17 \pm 0.19$ | $2.26 \pm 0.51$ |

Table 4: Comparison between our proposed method and other baselines on Photo, Computers and Arxiv in terms of node classification accuracy (Acc%) and expected calibration error (ECE%). The experiments are repeated 10 times and the average results and standard deviation are reported. The bold represents the best results.

| Methods | Amazon-Photo | | Amazon-Computers | | OGB-Arxiv | |
|---|---|---|---|---|---|---|
| | Acc | ECE | Acc | ECE | Acc | ECE |
| GCN (Kipf & Welling, 2016) | $93.30 \pm 0.72$ | $3.14 \pm 0.40$ | $88.23 \pm 0.44$ | $6.45 \pm 0.52$ | $80.38 \pm 0.48$ | $5.02 \pm 0.25$ |
| CaGCN (Wang et al., 2021b) | $91.73 \pm 0.96$ | $3.29 \pm 0.54$ | $87.74 \pm 0.51$ | $\mathbf{2.53} \pm 0.19$ | $\mathbf{80.44} \pm 0.42$ | $15.01 \pm 2.43$ |
| GATS (Hsu et al., 2022b) | $91.31 \pm 0.92$ | $3.79 \pm 1.89$ | $87.41 \pm 0.62$ | $3.75 \pm 0.63$ | $80.18 \pm 0.86$ | $4.06 \pm 0.46$ |
| GCL (Wang et al., 2022) | $93.17 \pm 0.58$ | $3.67 \pm 1.28$ | $87.53 \pm 1.37$ | $6.88 \pm 0.49$ | $80.35 \pm 0.51$ | $4.87 \pm 0.31$ |
| OODGAT (Song & Wang, 2022) | $90.53 \pm 0.66$ | $4.82 \pm 2.48$ | $88.23 \pm 0.63$ | $4.72 \pm 0.74$ | $71.36 \pm 1.53$ | $10.45 \pm 2.15$ |
| HyperU-GCN (Yang et al., 2022) | $92.16 \pm 1.39$ | $2.93 \pm 1.27$ | $\mathbf{89.53} \pm 1.05$ | $5.80 \pm 0.75$ | $78.28 \pm 1.76$ | $4.84 \pm 1.47$ |
| RNGER+GCN (Ours) | $\mathbf{93.55} \pm 0.79$ | $\mathbf{2.21} \pm 0.48$ | $87.92 \pm 0.76$ | $4.75 \pm 0.75$ | $79.92 \pm 0.45$ | $4.90 \pm 0.41$ |
| RNGER+GATS (Ours) | $93.45 \pm 0.83$ | $2.95 \pm 0.63$ | $87.39 \pm 0.66$ | $3.41 \pm 0.64$ | $79.95 \pm 0.78$ | $\mathbf{3.87} \pm 0.72$ |

& Welling, 2016). However, in some datasets in which over-confidence dominates, the GCL (Wang et al., 2022) would be less effective. OODGAT (Song & Wang, 2022) can identify the potential OOD nodes during the training and reduce the connection between the ID and OOD nodes by lowering the corresponding edge weights. However, our experimental results show that it would still suffer large calibration errors on some benchmark datasets.

Compared to the baselines, our method does not explicitly need to identify the OOD nodes. When wrapped with our framework, the existing GNN models have better performance on the ECE with comparable accuracy compared to their corresponding baseline methods. For instance, RNGER+GCN can achieve the ECE of 1.07% and 2.21% on PubMed (Yang et al., 2016) and Amazon-Photo (Shchur et al., 2018). RNGER+GATS can achieve best ECE result on Arxiv (Hu et al., 2020). Besides, RNGER+GCN also outperforms the GCN (Kipf & Welling, 2016) with original edge weights However, on some datasets our method is less effective in calibration of GNNs than some baselines on some datasets. Basically, our method is more effective on larger datasets.

To better visualize the ECE, the ECE for our method and the baselines On Cora (Yang et al., 2016) are illustrated in Fig. 3. Well-calibrated results is supposed to have closer alignment with the expected results along the diagonal line. The Fig. 3 demonstrates the better alignment of our method to the diagonal line than that of other baselines, which is consistent with our experimental results.

## 7   Conclusion

In this paper, we focus on the calibration of GNNs when graph is mixed with OOD nodes. When a graph is noisy, the noisy information from the OOD nodes would be propagated to the ID nodes, and existing calibration method is less effective on the noisy graph. To address this problem, we proposed the RL-enhanced Node-wise Graph Edge Re-weighting (RNGER) framework that aims to calibrate the graph neural network by the modified edge weights . Existing GNNs can be incorporated into our framework and extensive results on benchmarks demonstrate that our framework can calibrate the GNNs with the presence of OOD nodes and obtain comparable accuracy.

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

## A APPENDIX

### A.1 EXPERIMENTAL SETTING DETAILS

**Implementation Details**. **GCN** (Kipf & Welling, 2016): The learning rate is 1e-2, weight decay is 5e-4. The hidden dimension is 128 with two layers. We choose Adam(Kingma & Ba, 2014) optimizer to train the model.

**CaGCN** (Wang et al., 2021b) calibrates the confidence of the GNN by the post-hoc method to ensure the reliability of the prediction, with an estimation of different types of uncertainty. In the experiment we choose GCN (Kipf & Welling, 2016) as the base model. The hidden dimension is 128. The initial learning rate is 1e-2. The number of head is 8.

**GATS** (Hsu et al., 2022b): GATS proposed a new temperature scaling techniques to calibrate the graph neural networks. we choose GCN (Kipf & Welling, 2016) as the base model. The hidden dimension is 128. In the experiment. The weight decay is 5e-3. The number of head is 8 and the value of bias is 1.

**GCL** (Wang et al., 2022): GCL loss function is proposed to calibrate the graph neural network in an end-to-end manner. The coefficient $\gamma$ is set to 0.05. The hidden dimension is 128. The rest setting is the same as GCN (Kipf & Welling, 2016).

**OODGAT** (Song & Wang, 2022): OODGAT aims to perform the node classification and OOD detection simultaneously when graph is mixed with OOD nodes. We adopt the same experimental setting as the original work. Note that the ID/OOD split in our experiment is different from that of OODGAT.

**HyperU-GCN** (Yang et al., 2022) focuses on automated graph learning which can obtain the optimal hyperparameters through joint optimization on model weights and hyperparameters. We adopt the same experimental setting as the original work.

Table 5: Performance of RNGER w/o indicator function or entropy in the reward and of the complete reward. Average results of 10 runs are reported.

| Methods | Cora | | Citeseer | |
|---|---|---|---|---|
| | Acc | ECE | Acc | ECE |
| RNGER w/o entropy | **91.99** | 1.36 | 93.66 | 2.77 |
| RNGER w/o indicator | 91.04 | 1.10 | 91.63 | 3.83 |
| RNGER complete | 91.79 | **1.07** | 93.55 | **2.21** |

Table 6: The performance of GKDE-GCN on node classification and OOD node detection with old and new edge weights. The bold represents the best results.

| Dataset | Edge weight | Acc(%) | ECE(%) | OOD AUROC(%) | OOD AUPR(%) |
|---|---|---|---|---|---|
| PubMed | original | **85.61** | 10.73 | 85.21 | 73.58 |
| | Modified | 85.28 | **9.70** | **85.39** | **72.46** |
| Citeseer | original | 65.43 | 4.56 | 80.75 | 81.72 |
| | Modified | **67.93** | **3.56** | **83.41** | **83.57** |
| Amazon Photo | original | 89.83 | 3.05 | 69.05 | 61.35 |
| | Modified | **91.42** | **1.80** | **69.90** | **62.16** |

In our method, we adopt GCN (Kipf & Welling, 2016) and HyperU-GCN (Yang et al., 2022) as our GNN backbone. The hyper-parameters of GCN are the same as the corresponding baselines. The learning rate is 1e-2 and weight decay is 5e-4. The hidden dimension is 128. The Actor and Critic in our framework are implemented as a three-layered MLP with the dimension of hidden layers 256 and 16, respectively. Adam (Kingma & Ba, 2014) is adopted for training optimization with the learning rate of 1e-3 for Actor and Critic, The weight decay is 1e-2. The $\alpha$ is set to 0.95, and the discount coefficient $\gamma$ is 0.95. $\sigma_0$ is set to 0.1. The size of replay buffer is 1e4 and the total number of episode $P$ is 50. The training epoch is 600 for Arxiv and 400 for others. We adopt 10 bins for evaluation of expected calibration error. All the experiments are running on NVIDIA A5000. We test our method and baselines 10 times with different seeds and the average results are reported.

## A.2 ABLATION STUDY

In our framework, the reward consists two terms: indicator function and entropy. To investigate the contribution of each term in the reward, we conduct an experiment in which only one term is available in the reward. The experiments are repeated 10 times on PubMed and Amazon Photo and the average results are reported. The results are shown in Table 5. Indicator function can ensure the comparable accuracy achieved by the model. Without an indicator function, the model may deviate from the correct prediction by simply enlarging/lowering the entropy. Without entropy in the reward, our method would be less effective to calibrate the graph neural networks.

## A.3 CASE STUDY

We conduct a case study to investigate if the adjusted edge weights can improve the graph learning performance of other methods. GKDE-GCN (Zhao et al., 2020) is a representative method for detecting out-of-distribution (OOD) nodes by uncertainty estimation. We evaluate the performance of GKDE-GCN (Zhao et al., 2020) on node classification and OOD node detection with the adjusted edge weights learned by our framework. The metrics for the OOD node detection are AUROC and AUPR. We run the experiments 10 times on Cora, Citeseer, and PubMed (Yang et al., 2016), and report the average results in Table 6. The results show that our adjusted edge weights can help improve the node classification and OOD detection performance of the base model GKDE-GCN (Zhao et al., 2020).

