# OpenReview forum: "Node-wise Calibration of Graph Neural Networks under Out-of-Distribution Nodes via Reinforcement Learning"
_ICLR.cc/2024/Conference — ICLR 2024 Conference Withdrawn Submission_

### Official Review · Reviewer_mT4R · 2023-10-23

**Soundness:** 2 fair
**Presentation:** 3 good
**Contribution:** 2 fair
**Rating:** 3
**Confidence:** 3

**Summary:**

This paper addresses the calibration issue in node classification. The authors show that GNNs are poorly calibrated when graphs have OOD nodes and lowering the weights of the edges connecting to OOD nodes can improve the calibration. Based on these observations, they propose a method to identify these edges and reweight them automatically through DDPG. Specifically, they design state, action, and reward as edge features, edge weights, and the weighted sum of accuracy and carefully designed entropy. Then, they iteratively sample edges from self-loops to other edges within 2-hop subgraphs while computing reward at each step. In experiments, they show that their algorithm can improve calibration in various datasets and detect OOD nodes well.

**Strengths:**

* The authors empirically demonstrate that the edges connecting to OOD nodes might harm the calibration.
* They propose the algorithm to be able to identify these edges and determine weights automatically via DDPG.

**Weaknesses:**

* (W1) The analysis of poor calibration in graphs with OOD nodes seems insufficient. I cannot find why the edges connecting to OOD nodes can harm the calibration (The authors only provide the empirical results without analysis). It would be better to provide a detailed explanation about it.
* (W2) The performance gain (ECE) including (RNGER+GCN v.s. GCN, and RNGER+GATS v.s. GATS) seems marginal in 4 out of 6 datasets such as Cora, Citeseer, PubMed, and OGB-Arxiv in Table 1 and 2.
* (W3) The proposed method seems to require many iterations to train models under dense and large graphs in that models learn the appropriate edge weight for all edges.
* (W4) The used graphs in experiments seem not diverse. It would be better to include heterophilous graphs.

Typos
* Underscript of loss in Proposition 2: L_inv_FL
* The time complexity: O(N_d*(L|E|F+)d+LNF^2+Fh)
* The bold is wrong in Table 6 (PubMed, OOD AUPR).

**Questions:**

* (related to W3) Could you compare the full training costs compared to baselines and naive GNN?
* Why does the proposed algorithm perform significantly well on AmazonPhoto and AmazonComputers compared to other datasets?

---

### Official Review · Reviewer_YYpv · 2023-10-31

**Soundness:** 2 fair
**Presentation:** 3 good
**Contribution:** 2 fair
**Rating:** 3
**Confidence:** 4

**Summary:**

This paper studies node-wise calibration of GNNs under OOD nodes via RL. The proposed method adjusts the weights of edges to address the calibration issue. The proposed method RNGER calibrate GNNs against OOD nodes and explores the entropy of targets and the adjustment of edge weights without the need of identifying OOD nodes. Experiments are conducted on benchmark datasets.

**Strengths:**

1.	The problem studied in this paper is important and useful in the GNNs.
2.	The idea of adjusting edge weights is reasonable to handle the issue.
3.	It is with sufficient empirical studies to clearly explain the motivation and idea.

**Weaknesses:**

1.	A major concern is in Table 3 and 4, experimental results. The proposed RNGER is with inferior Acc and ECE, compared with existing methods, under most datasets. Therefore, I think the effectiveness of the proposed method is not good enough.
2.	In section 5, methodology, the proposed method is motivated by many recent methods, and the design of using RL is straightforward, making the novelty and contribution of the paper unclear. Also, it is not challenging to design the method.
3.	In the paper, related work and background sections take up too much space, which can be improved.

**Questions:**

Please see the weaknesses above

---

### Official Review · Reviewer_cXQX · 2023-10-31

**Soundness:** 2 fair
**Presentation:** 2 fair
**Contribution:** 2 fair
**Rating:** 3
**Confidence:** 4

**Summary:**

The paper proposes an RL-enhanced Node-wise Graph Edge Re-weighting (RNGER) framework to address the calibration issue of Graph Neural Networks (GNNs) when dealing with graphs that contain out-of-distribution (OOD) nodes. The existing calibration methods are less effective on graphs with OOD nodes as they do not consider the negative impact of OOD nodes. The proposed framework incorporates reinforcement learning (RL) to learn new edge weights that can adjust the entropy of the output from GNNs and mitigate the calibration issue. Experimental results demonstrate that the RNGER framework can effectively calibrate GNNs and achieve comparable accuracy compared to state-of-the-art methods. The learned edge weights are also transferable and beneficial in other graph learning tasks.

**Strengths:**

1. The structure of this paper is clear and easy to understand.
2. The negative impact of OOD nodes in GNN is a good problem that has practical application.
3. Various experiments are conducted and the results are well analyzed.

**Weaknesses:**

1. The time complexity is confusing. Other typos that I find out will be pointed out in Q1.
2. Empirical experiments are carried out when all OOD nodes are known, and the effect is not obvious in the case of unknown OOD, which leads to insufficient motivation.
3. As stated in Intro and Related Work, there are some previous works focus on OOD nodes. The shortcomings of existing works are not well discussed in this paper, and the experimental results do not show significantly better results than existing works(e.g. On Cora, Citeseer and Amazon-Computers, baselines outperform the proposed method).
4. Proposed algorithm needs to compute 2-hop edges, which makes the time complexity unacceptable and limits its potenial for real-world large graphs.
5. Algorithm 1 is confused since it contains too many words without equations and misses output.

**Questions:**

1. In Section 1, '... the existing calibration method would be less effective on the graph ...' should be '... the existingcalibration methods would be less effective on the graphs ...', 'Inspired by the previous work' should be ' ... previous works', '... implicitly adjustment of the entropy' should be 'implicit adjustment of the entropy', 'Existing GNN ...' should be 'Existing GNNs'. In Section 3.1, all the intersection of sets should be the union of sets. All matrices should be bold and black. The time complexity is confusing, please rewrite it.
2. Can you provide analysis on the effect of edge-reweighting without considering specific OOD nodes?

---

### Official Review · Reviewer_Tjfh · 2023-11-02

**Soundness:** 3 good
**Presentation:** 3 good
**Contribution:** 3 good
**Rating:** 6
**Confidence:** 4

**Summary:**

The paper aims to address the problem of calibration of GNNs with out-of-distribution (OOD) nodes. The empirical study suggested that the adjusted edge weights can lower the calibration error, and based on this finding, the authors proposed an RL-based edge reweighting method to reduce the calibration error with modified edge weights. Experimental results show that the proposed method achieves promising results on some benchmarks.

**Strengths:**

1.	The problem discussed in the paper is new and significant. Investigating the calibration of GNNs with OOD nodes is a timely and important research problem. The finding that GNNs are either over-confident or under-confident on different benchmarks is quite interesting.

2.	The proposed method is well-motivated, and the empirical experiments provide a sound foundation for the proposed method. The proposed method can successfully reduce the calibration error of GNNs by the implicit regularization of entropy through the adjusted edge weights.

3.	The proposed method is sufficiently evaluated on benchmark datasets with different sizes. The experimental results show that the proposed method can achieve promising performance on benchmark datasets.

4.	The paper is well-organized and clearly written. Technical details are easy to follow.

**Weaknesses:**

1.	For the propositions in this paper, the authors should provide detailed proof, which will be very helpful for readers who are interested in theoretical analysis of this topic.

2.	More justifications and analyses should be provided in the experiments. For example, the author can provide a visual illustration of the distribution of the updated edge weights.

3.	Current experiments don’t include the result of RNGER+CaGCN. Does the method also achieve better calibration performance when it’s incorporated with CaGCN?

**Questions:**

1.	The sampling is 2 hops. Why is the step of hop fixed at 2?

2.	Is the proposed method sensitive to the different split of ID and OOD nodes in the benchmark datasets?

3.	What is the performance when CaGCN is incorporated into the proposed method?

4.	What is the distribution of the modified edge weights obtained from the proposed method?